# Continuous Dictionary of Nodes Model and Bilinear-Diffusion Representation Learning for Brain Disease Analysis

**DOI:** 10.3390/brainsci14080810

**Published:** 2024-08-13

**Authors:** Jiarui Liang, Tianyi Yan, Yin Huang, Ting Li, Songhui Rao, Hongye Yang, Jiayu Lu, Yan Niu, Dandan Li, Jie Xiang, Bin Wang

**Affiliations:** 1School of Computer Science and Technology (School of Data Science), Taiyuan University of Technology, Taiyuan 030024, China; 2School of Medical Technology, Beijing Institute of Technology, Beijing 100081, China; yantianyi@bit.edu.cn

**Keywords:** brain diseases, brain network, schizophrenia, bipolar disorder, functional magnetic resonance imaging (fMRI), representation learning

## Abstract

Brain networks based on functional magnetic resonance imaging (fMRI) provide a crucial perspective for diagnosing brain diseases. Representation learning has recently attracted tremendous attention due to its strong representation capability, which can be naturally applied to brain disease analysis. However, traditional representation learning only considers direct and local node interactions in original brain networks, posing challenges in constructing higher-order brain networks to represent indirect and extensive node interactions. To address this problem, we propose the Continuous Dictionary of Nodes model and Bilinear-Diffusion (CDON-BD) network for brain disease analysis. The CDON model is innovatively used to learn the original brain network, with its encoder weights directly regarded as latent features. To fully integrate latent features, we further utilize Bilinear Pooling to construct higher-order brain networks. The Diffusion Module is designed to capture extensive node interactions in higher-order brain networks. Compared to state-of-the-art methods, CDON-BD demonstrates competitive classification performance on two real datasets. Moreover, the higher-order representations learned by our method reveal brain regions relevant to the diseases, contributing to a better understanding of the pathology of brain diseases.

## 1. Introduction

Various neuroimaging techniques have been widely applied in the research and analysis of brain diseases [1,2]. Functional Magnetic Resonance Imaging (fMRI) is one of the most commonly used neuroimaging techniques that captures blood-oxygen-level-dependent (BOLD) signals from various brain regions [3,4]. It then analyzes the correlation of BOLD signals between different brain regions to construct brain networks [5]. The brain network consists of nodes and connections, where nodes represent brain regions, and connections represent the physiological correlations between brain regions [6]. Through the analysis of the brain network, we can gain insights into the functional organization and information transmission pathways of the brain, deepening our understanding of the inner workings of the brain and the mechanisms underlying related diseases [7,8]. Additionally, brain networks provide a powerful tool for diagnosing brain diseases [9].

Schizophrenia (SZ) and bipolar disorder (BD) are both severe mental illnesses. Connectivity impairments between brain regions in SZ and BD have been observed and demonstrated through various neuroimaging techniques, such as fMRI [10,11,12]. SZ and BD are pathological conditions that show a series of cognitive, emotional, and behavioral alterations [13,14]. Several studies have found that SZ and BD show network alterations compared to healthy controls (HC) [15,16,17], with potential widespread connectivity disruptions between brain regions (e.g., reduced connections between the frontal and temporal lobe white matter) [18,19]. However, most research focuses on the direct relationships between brain regions, neglecting the higher-order relationships implied by indirect connections [20], which limits the performance of disease diagnosis [21,22,23]. Previous studies have shown that the latent features in indirect connections may be closely related to the disease and are crucial for mapping the higher-order relationships of brain networks [24,25]. Additionally, the overlapping and similar symptoms of SZ and BD, such as cognitive difficulties and emotional abnormalities, further limit the exploration and analysis of distinguishing different diseases [26]. Therefore, constructing higher-order relationships in the brain networks of schizophrenia and bipolar disorder is necessary.

On the other hand, traditional representation learning methods for brain networks typically focus on local node interactions, often neglecting the extensive node interactions [27,28]. These methods prioritize analyzing direct, localized connections between pairwise brain regions [24,29]; still, they cannot capture broader, extensive connections involving three or more brain regions that are essential for comprehensive brain network analysis [6]. Recent studies have demonstrated that brain function involves a widely connected complex network rather than being limited to localized regions [30,31], highlighting the critical importance of considering extensive node interactions in brain disease analysis. To capture extensive node interactions, diffusion methods based on random walks [32] have gained popularity. Random walks take into account the relationship between nodes, allowing information to diffuse from the current node to its neighboring or related nodes through connections [33]. During the diffusion process, features spread from the selected nodes to all their connected nodes, thereby learning more comprehensive and extensive node representations [34]. Therefore, diffusion processes based on random walks hold tremendous potential in capturing the extensive node interactions within brain networks, allowing for a more comprehensive representation of the brain.

To address the above problems, in this paper, we propose the Continuous Dictionary of Nodes model and Bilinear-Diffusion (CDON-BD) network to learn the representations for brain disease analysis. The Continuous Dictionary of Nodes (CDON) model directly captures the latent features of the brain network through encoder weights, avoiding the potential information loss of brain functional connectivity. Bilinear Pooling is subsequently employed to construct higher-order brain networks. To capture extensive node interactions, we introduced a Diffusion Module designed to learn the higher-order brain network (obtained by the CDON and Bilinear Pooling) and generate higher-order representations for disease diagnosis. The framework of the CDON-BD is shown in Figure 1. Specifically, we first extract time series from fMRI and construct brain networks. Then, the brain network is split column-by-column and fed into CDON to capture latent features. Subsequently, latent features are fed into a Bilinear Pooling layer to obtain the higher-order brain network. Based on this, the Diffusion Module captures extensive node interactions in an unsupervised manner. Ultimately, CDON-BD automatically generates higher-order representations from higher-order brain networks for disease analysis.

Experiments on real datasets demonstrate the effectiveness of our method. Compared to state-of-the-art brain disease analysis methods, CDON-BD demonstrates competitive performance. The main contributions of this paper can be summarized as follows:A new Continuous Dictionary of Nodes (CDON) model captures the latent features of brain networks through its encoder weights. CDON efficiently and directly captures brain functional connections through parameter weights, avoiding potential information loss in brain disease analysis.The Bilinear Pooling technique is innovatively employed to construct higher-order brain networks. Higher-order brain networks integrate latent features, allowing for a global representation of the brain.A novel Diffusion Module captures extensive node interactions in higher-order brain networks, generating higher-order representations for disease diagnosis in an unsupervised manner.

## 2. Related Works

This section systematically reviews disease analyses of brain networks with latent features. Then, related works on representation learning methods for graph-structured data are summarized.

### 2.1. Brain Networks with Latent Features for Disease Analysis

Brain networks are usually functional or structural networks present in the brain and may be altered by pathological conditions [15,16,17]. Latent features in indirect connections of brain networks may be closely related to diseases [35]. Niu et al. [36] utilized functional entropy to extract latent features of brain networks in patients with schizophrenia and bipolar disorder at the global, modular, and nodal levels. Additionally, Liu et al. [37] utilized a cascaded convolutional neural network (CNN) to learn hierarchical and latent features of brain networks for Alzheimer’s disease classification. Masoudi et al. [38] proposed a 3D CNN that integrates multimodal information to generate higher-order representations for the diagnosis of schizophrenia. Although CNNs can effectively extract latent features from brain networks [21,22], they result in significant computational costs. It is noteworthy that Autoencoder [23], as a baseline method, can learn latent features of brain networks through an encoder–decoder architecture, but it is insensitive to disease classification. Although these methods can extract latent features from the original brain networks, they have not fully utilized these features to construct higher-order brain networks. Higher-order brain networks often contain richer latent features, thus enabling better understanding and diagnosis of brain diseases [20]. Therefore, we propose the Continuous Dictionary of Nodes (CDON) model combined with the Bilinear Pooling technique to capture latent features and construct higher-order brain networks. CDON directly treats encoder weights as latent features, thus avoiding potential information loss in brain disease analysis.

### 2.2. Representation Learning for Graph-Structured Data

In recent years, representation learning methods for graph-structured data have gained tremendous popularity. In brain network analysis, representation learning can efficiently capture network and node representations [39,40], thus enhancing the diagnostic performance of brain diseases. Huang et al. [27] proposed a node-level structural embedding and alignment representation learning framework (nSEAL) for representing brain networks at the node level. Shi et al. [29] employed graph neural networks and introduced a heterogeneous graph neural network (HebrainGNN) to learn brain networks. Meanwhile, Liu et al. [24] developed an enhanced multi-modal graph convolutional network (MME-GCN) that integrates structural and functional brain graphs for disease classification. Additionally, Chen et al. [28] proposed an orthogonal latent feature graph (OLFG) model with feature weighting and representation learning. OLFG achieves disease diagnosis by learning representations of brain networks based on graphs and spatial information. However, existing representation learning methods only consider interactions of local nodes, neglecting extensive node interactions. Therefore, we utilize a Diffusion Module to capture extensive node interactions in the brain network, generating higher-order representations for diagnosing brain diseases.

## 3. Materials and Methods

This section first describes the acquisition and preprocessing of fMRI data. Following this, it introduces the Continuous Dictionary of Nodes Model, which captures latent features in brain networks. Next, the Bilinear Pooling technique is introduced for constructing higher-order brain networks. Then, the process of generating higher-order representations using the Diffusion Module is described. Finally, the classification methods and evaluation metrics are presented.

### 3.1. Data Acquisition and Processing

Resting-state fMRI data were obtained from the University of California, Los Angeles (UCLA) Neuropsychiatric Phenomics Consortium LA5c study, including 50 healthy controls (HC), 48 patients with schizophrenia (SZ), and 49 patients with bipolar disorder (BD). All participants were between 21 and 50 years of age, with no differences in age or gender distribution. For further detailed demographic information, please refer to Table 1.

The fMRI data were preprocessed by using DPABI [41]. The preprocessing procedure consists of several steps: removal of the first ten time points, slice time correction, motion correction (with the first image serving as the reference), registration to a 4 × 4 × 4 voxel resolution using the Montreal Neurological Institute (MNI) [42] space, spatial smoothing with a 4 mm full-width half-maximum (FWHM) Gaussian kernel [20], and removal of low-frequency drift and high-frequency noise through linear detrending and bandpass filtering (0.01–0.25 Hz) [43].

The brains are segmented into regions of interest (ROI) using the Automated Anatomical Labeling (AAL) [42] atlas. Brain networks are constructed by calculating the Pearson correlation coefficient between the regional BOLD signals. Finally, Fisher’s r-to-z transformation is applied to brain networks to normalize their sampling distribution of correlation coefficients.

### 3.2. Continuous Dictionary of Nodes Model

As shown in Figure 1, we introduce a CDON-BD network to learn higher-order representations from brain networks for disease diagnosis. In the framework of our method, a CDON model is employed for each participant to capture latent features.

Before training CDON to capture latent features, we apply a transformation operation to the brain network. The transformation operation splits the brain network into columns. Specifically, each participant’s brain network (90 × 90) is divided into 90 samples, each of which is a 90 × 1 feature vector. The benefit of splitting the brain network is that it allows CDON to learn the features of each node.

After the transformation, CDON learned the feature vectors node by node. To reduce computational costs, we designed a three-layer linear autoencoder consisting of input, hidden, and output layers. The number of units in the input and output layers matches the feature vector, whereas the number of units in the hidden layer is less than the feature vector. The training process of CDON is defined as follows:(1)$$\begin{array}{cccc} & & h=g_{\theta_{e}}\left(x\right)=\sigma \left(W_{e}x+b_{e}\right), & \end{array}$$
(2)$$\begin{array}{cccc} & & \hat{x}=g_{\theta_{d}}\left(h\right)=\sigma \left(W_{d}h+b_{d}\right), & \end{array}$$
where *x* is the feature vector, $x=\left\{x_{1},x_{2},\ldots x_{n}\right\},x_{i}\in R^{n\times 1}$, and $\hat{x}$ is the output of CDON, $\hat{x}=\left\{\hat{x_{1}},\hat{x_{2}},\ldots \hat{x_{n}}\right\},\hat{x_{i}}\in R^{n\times 1}$. *h* represents the hidden layer, $h=\left\{h_{1},h_{2},\ldots h_{n}\right\},h_{i}\in R^{m\times 1}$. $W_{e}\in R^{m\times 1}$ and $W_{d}\in R^{m\times 1}$ represent the weights of the encoder and decoder, respectively. The optimization objective function of CDON is defined as
(3)$$\begin{array}{cccc} & & MinimizeLoss=dist(x,\hat{x}) & \end{array}$$

The encoder weights $W_{e}$ can automatically capture latent features, enabling the extraction of deep information from the data [44]. The innovation of CDON lies in treating encoder weights as latent features, directly capturing brain functional connections, and avoiding potential information loss. Compared to traditional methods, this is a more efficient and direct feature extraction approach without the need for a test set.

### 3.3. Constructing Higher-Order Brain Networks via Bilinear Pooling

As shown in Figure 1, the latent features are captured by the encoder weights of CDON, where each row represents the features of a node. Currently, these features are limited to the node level and lack explicit representation of pairwise correlations between nodes. Therefore, there is a significant necessity to fuse the latent features, which can represent the brain network from a global perspective.

Bilinear Pooling is a classical feature fusion method that has been proven to be effective in fusing features at the node level. Huang et al. [45] use Bilinear Pooling to extract the second-order statistics of each node representation, which fuses node-level latent features. The network-level features based on connectivity demonstrate better performance in the disease classification task. Based on Bilinear Pooling, latent features can be extended from the node to the network. Hence, we use the Bilinear Pooling technique to construct higher-order brain networks. The higher-order brain network is defined as
(4)$$\begin{array}{cccc} & & B=W_{e}\cdot {W_{e}}^{T}, & \end{array}$$
where $B\in R^{n\times n}$ represents the higher-order brain network. *B* is calculated by homologous Bilinear Pooling, i.e., the inner product of the latent features. Higher-order brain networks capture pairwise correlations of nodes, fusing node-level latent features. The perspective of the whole brain representation is elevated from the local to the global.

### 3.4. Diffusion Module

The primary goal of our work is to obtain higher-order representations. In our method, we design a Diffusion Module applied to higher-order brain networks and learn higher-order representations. The Diffusion Module consists of two main parts: a diffusion layer and a continuous bag-of-words (CBOW) [46] layer.

We employ a breadth-first random walk strategy to diffuse among nodes in the diffusion layer. The advantage of this approach is that it enables the discovery of the shortest path between nodes and helps avoid getting trapped in local optimal solutions. Random walks on higher-order brain networks can generate sequences of nodes. Specifically, the random walk strategy takes into account the relationship between the current node and its neighboring nodes to determine the next step, and the diffusion layer introduces parameters *p* and *q* to control the strategy.
(5)$$\begin{array}{cccc} & & \lambda \left(s,t\right)=\left\{\begin{array}{ccc} 1/p, & & \mathrm{if}\;d_{st}=0\\ 1, & & \mathrm{if}\;d_{st}=1\\ 1/q, & & \mathrm{if}\;d_{st}=2 \end{array}\right., & \end{array}$$
(6)$$\begin{array}{cccc} & & p\left(c,t\right)=\lambda \left(s,t\right)\cdot \omega_{ct}, & \end{array}$$
where *c* denotes the current node after the random walk through the edge (s,c), and *t* denotes the target node for the next step. The transfer probability of the random walk follows the strategy of Equation (6), λ(s,t) denotes the weights of the connectome edges, $\omega_{ct}$ denotes the weights of the edges between nodes *c* and *t*, and $d_{st}$ denotes the distance of the shortest path from node *s* to target node *t*.

After diffusion, the generated sequences of nodes are fed into the CBOW layer. The continuous bag-of-words model generates higher-order representations by regarding the node sequences as word sequences. Each node in the node sequence is represented as a one-hot encoding. The objective of the CBOW model is to maximize the conditional probability of predicting the target node given the neighboring nodes. Hence, the objective function of the CBOW layer is defined as
(7)$$\begin{array}{cccc} & & \tau =\sum_{t=1}^{T}\log P(v_{t}|Nbhd\left(v_{t}\right)), & \end{array}$$
where *T* is the total number of nodes in the node sequence, $v_{t}$ is the target node, and $Nbhd(v_{t})$ represents the set of neighborhood nodes. To expedite training and circumvent the computation of SoftMax probabilities for all nodes, we adopt negative sampling as an efficient approximation technique [47]. The CBOW layer employs a neural network to learn higher-order representations and leverages backpropagation and gradient descent for parameter optimization, leading to improved higher-order representations.

### 3.5. Classification

To facilitate effective classification, we introduced brain templates. In brain research, the construction of brain templates tailored to specific domains, populations, and diseases is a common approach. Through brain templates, individual brain networks can be mapped to a common standardized space, reducing individual differences. Brain templates may capture some universal features of brain structure, thereby achieving dimensionality reduction to some extent. Wilke et al. [48] generated specific brain templates for analysis within age groups in the pediatric population. Fonov et al. [49] proposed an unbiased, age-appropriate method for constructing brain templates, enabling comparisons between studies within a pediatric-specific standardized space. Ashburner et al. [50] generated tissue probability maps that represent the average shape of brains across many participants, assessing the match between individual participants and the brain template in a public space. Additionally, there are studies on symmetrical templates for investigating brain hemisphere differences [51] and single-gender templates for studying gender disparities [52].

Based on this, we propose aligning individual higher-order representations using the average brain template and then calculating the network distance between individual higher-order representations and the template. Network distances can accurately capture network-level differences and hold great potential for disease prediction. Firstly, we establish the healthy brain template and the patient brain template denoted by $C^{+}$ and $C^{-}$, respectively:(8)$$\begin{array}{cccc} & & C^{+}=\frac{1}{m^{+}}\sum_{i=1}^{m^{+}}c_{i}^{+}, & \end{array}$$
(9)$$\begin{array}{cccc} & & C^{-}=\frac{1}{m^{-}}\sum_{i=1}^{m^{-}}c_{i}^{-}, & \end{array}$$
where, $m^{+}$ and $m^{-}$ represent the number of healthy individuals and patients, respectively. Based on these templates, we further proposed the network distance matrix $H\in R^{(m^{+}+m^{-})\times 2}$ to reflect the network distance between the target network and the reference template. For instance, the network distance between the target network *n* and two templates can be computed as
(10)$$\begin{array}{cccc} & & H_{\left((n,1)\right)}=CosDist\left(A_{n}^{\prime },C^{+}\right), & \end{array}$$
(11)$$\begin{array}{cccc} & & H_{\left((n,2)\right)}=CosDist\left(A_{n}^{\prime },C^{-}\right), & \end{array}$$
(12)$$\begin{array}{cccc} & & CosDist\left(a,b\right)=1-\cos\left(a,b\right)=\frac{\left||a\right|{|}_{2}\cdot \left||b\right|{|}_{2}-a\cdot b}{\left||a\right|{|}_{2}\cdot \left||b\right|{|}_{2}}, & \end{array}$$
where, CosDist(a,b) represents the cosine distance between the two representations of $a=(a_{1},a_{2},\ldots a_{t})$ and $b=(b_{1},b_{2},\ldots b_{n})$ which ranges from [0, 2].

Through the above methods, each sample obtained two types of network distances, corresponding to the healthy template and the patient template, respectively. Further, a Radial Basis Function Support Vector Machine (RBF-SVM) classifier was trained using these network distances. The classification performance was assessed through 10-fold cross-validation and four metrics: accuracy (ACC), sensitivity (SEN), specificity (SPE), and area under the receiver operating characteristic curve (AUC). The definitions of these metrics are provided below:(13)$$\begin{array}{cccc} & & ACC=\frac{TP+TN}{TP+FP+TN+FN},\;SEN=\frac{TP}{TP+FN},\;SPE=\frac{TN}{TN+FP}, & \end{array}$$
where TP, TN, FP, and FN represent true positive, true negative, false positive, and false negative values, respectively. AUC denotes the area under the receiver operating characteristic curve (ROC), and a larger AUC indicates better classifier performance.

The proposed CDON-BD is expected to distinguish patients from healthy controls and has the potential to be extended to distinguish between different diseases (SZ vs. BD). Additionally, we hope to validate the generalization ability of CDON-BD by conducting experiments and analyses on another dataset (COBRE). These experiments will validate the broad applicability and generalization of CDON-BD across various classification tasks and datasets.

## 4. Experiments and Results

In this section, we conducted several experiments to evaluate the performance of our method. Initially, using higher-order representations obtained through CDON-BD, we calculated both network distances and node distances. Then, we evaluated the classification performance of network distances on real datasets. For brain disease analysis, we assessed group differences based on node distances. In addition, we validated the generalizability of our method on another dataset, COBRE. Finally, we analyzed the impact of different input dimensions and latent features on the performance of CDON.

### 4.1. Classification Performance

We validated the classification performance of our method on a real dataset using RBF-SVM as the classifier. The primary task of the experiments is to classify healthy controls and patients, i.e., SZ vs. HC and BD vs. HC. Table 2 presents a comparison of the classification performance between the SOTA methods and our method.

In the SZ vs. HC classification task, our method achieved competitive performance, with an accuracy of 98.8% (SEN: 97.3%, SPE: 99.1%, AUC: 0.993). Similarly, in the BD vs. HC classification task, our method also demonstrated strong performance, achieving an accuracy of 98.5% (SEN: 98.6%, SPE: 97.8%, AUC: 0.980). Among the various competitive methods, OLFG [28] achieved the highest performance, with accuracies of 96.8% (in SZ vs. HC) and 96.7% (in BD vs. HC), albeit at least 1.8% lower than CDON-BD. It is noteworthy that the accuracy of CDON-BD significantly surpassed that of the baseline method (Autoencoder [23]), further validating the effectiveness of our proposed approach. Additionally, we observed excellent performance from CNN-based models [21,22,37], with 3D-CNN [38] achieving accuracies of 96.1% (in SZ vs. HC) and 96.0% (in BD vs. HC), respectively. Multi-kernel SVM [35] showed promising performance in both classification tasks, but it still lagged behind many deep learning methods. HebrainGNN [29] and MME-GCN [24] adopted graph neural networks, achieving a maximum classification accuracy of 96.0%, which was still at least 2.5% lower than CDON-BD. Even though GNN considers interactions between pairwise nodes, it is unable to capture more extensive interactions involving three or more nodes. Compared to SOTA methods, our proposed method yields remarkable results and holds promise for improving the diagnostic accuracy of brain diseases.

Schizophrenia and bipolar disorder share some common symptoms [26], such as mood swings and cognitive impairments, often requiring repeated clinical diagnoses [53]. However, existing studies have demonstrated the feasibility of classifying SZ and BP through brain networks, suggesting that differences in network structure and functional connectivity may become future diagnostic indicators [54,55,56]. To further evaluate the clinical applicability and sensitivity of CDON-BD to different diseases, we conducted SZ vs. BD classification experiments without considering healthy controls. Our method exhibited impressive performance in the SZ vs. BD classification task (see Figure 2), achieving an accuracy of 96.7% (SEN: 95.9%, SPE: 98.0%, AUC: 0.963), outperforming the latest brain network analysis methods by at least 6.7% [57,58]. It is worth noting that, possibly due to the similarity in symptoms between the two diseases, the model’s accuracy in the SZ vs. BD classification task is slightly lower than in the SZ vs. HC and BD vs. HC tasks. In future research, we will continue to explore the differences in brain networks between SZ and BD to improve accuracy and utility in practical diagnosis.

### 4.2. Analysis of Node Distances

In this subsection, we computed the node distances between the higher-order representations and the three types of brain templates: the SZ (schizophrenia) template, the BD (bipolar disorder) template, and the HC (health control) template. The calculation method for node distance is similar to network distance.

Based on the node distances, we further employ statistical analyses to precisely identify the specific brain regions showing significant group differences between patients and healthy controls. We perform a two-sample T-test (corrected using False Discovery Rate) on the node distances between healthy controls and patients, revealing brain regions with statistical differences. Figure 3 displays the brain regions with significant group differences (*p*-FDR < 0.05). To enhance the visualization of brain regions with significant differences, we applied a nonlinear transformation to the *p*-FDR, converting them into −log(p). As the *p*-FDR decreases, the corresponding −log(p) increases, leading to more pronounced displays in the figure.

Figure 3 displays a high degree of overlap between the HC template and the patient template, revealing several brain regions that exhibit significant differences in both templates. As shown in Figure 3a, there are significant differences in brain regions between healthy controls and individuals with schizophrenia, primarily concentrated in the Middle Frontal Gyrus, Postcentral Gyrus, Cingulate Gyrus, Precuneus, and Cuneus. Lesions in these areas may lead to abnormalities in the language center, visual center, and sensory center. As can be seen in Figure 3b, there are significant differences in brain regions between healthy controls and individuals with bipolar disorder, mainly distributed in the Inferior Frontal Gyrus, Middle Frontal Gyrus, and Paracentral Lobule, potentially resulting in abnormalities in the motor center and language center in patients. Our approach has the potential to uncover brain regions that may be associated with schizophrenia and bipolar disorder.

### 4.3. Classification Results on COBRE Dataset

In this section, we further validated the effectiveness of our proposed method using The Center for Biomedical Research Excellence (COBRE) dataset. We collected resting-state fMRI data from 37 healthy controls (HC) and 37 patients with schizophrenia and applied the aforementioned preprocessing methods. There were no significant differences in phenotypic information such as age and gender of the subjects. Detailed phenotypic information of the subjects is available on the COBRE website. Table 3 reports the classification performance of different methods on the COBRE dataset. Our proposed method achieved excellent performance in the SZ vs. HC classification task, with a classification accuracy of 98.6%. Although 3D CNN [59] scored higher on specificity, CDON-BD outperformed 3D CNN comprehensively in terms of accuracy and sensitivity. Furthermore, we observed that the performance of most deep learning methods surpassed that of machine learning methods, indicating the superior capability of neural networks in extracting features within brain networks. It is noteworthy that the baseline method (Autoencoder [23]) achieved only 73.6% classification accuracy, much lower than most methods. One possible reason is that conventional autoencoders are not suited to brain disease analysis, further highlighting the effectiveness of our proposed improvements.

### 4.4. Analysis of Continuous Dictionary of Nodes Model

To delve deeper into the Continuous Dictionary of Nodes (CDON) model, we analyzed the impact of input dimensions on the results. In our method, to match the 90 × 90 brain network, we set the input dimension of CDON to 90. Consequently, it is crucial to explore various dimensions to identify the optimal input setting for CDON. Figure 4 illustrates that CDON’s performance experiences a decline with an increase in the input dimension. Notably, when the input dimension reaches 4005 (90 × 89/2, the brain network is a symmetric matrix), CDON displays the lowest classification performance. Hence, we determine that the optimal input dimension for CDON is 90. More significantly, a dimension of 90 aligns with the 90 regions of interest (ROI) in the AAL template, holding considerable biological significance.

In addition, CDON captures two types of latent features ($W_{e}$ and $W_{d}$), and we also need to evaluate how these two types of latent features affect the results. In the optimal input dimension, we compared the effect of different features on performance (Figure 4). When selecting $W_{e}$ as the latent features, CDON achieved higher performance. Experimental results show that $W_{e}$ are the better latent features.

## 5. Discussion

In the discussion, we first demonstrated the effectiveness of Bilinear Pooling and the Diffusion Module. Then, the effects of the number of units in the hidden layer and the dimension of the higher-order representations obtained from the Diffusion Module on accuracy were evaluated. Next, we analyzed the effect of various latent features ($W_{e}$ and $W_{d}$) on performance. Finally, the limitations and future work of this study are discussed.

### 5.1. Effectiveness of the Bilinear Pooling and the Diffusion Module

The Bilinear Pooling technique and the Diffusion Module play a crucial role and make a substantial contribution to the performance. To demonstrate the role of the Bilinear Pooling and the Diffusion Module, we performed several ablation experiments. As depicted in Figure 5a, the integration of CDON with Bilinear Pooling (CDON-B) can enhance classification performance by at least 3.6% for both SZ vs. HC and BD vs. HC comparisons. Furthermore, the combination of CDON with Bilinear Pooling and the Diffusion Module (CDON-BD) demonstrates an improvement in classification performance by at least 7.5% for both SZ vs. HC and BD vs. HC comparisons. This highlights the importance of the Bilinear Pooling for fusing latent features and the Diffusion Module for capturing extensive node interactions.

### 5.2. Hidden Layer of CDON

Capturing features through the encoder weights of CDON has been found to substantially enhance the performance. Therefore, investigating the hyperparameters of CDON becomes essential. Among these hyperparameters, the number of hidden layers and the number of units play crucial roles in determining the performance of the model.

Taking into account the dataset size and the physiological significance of latent features, we have designed a single hidden layer architecture for CDON. By employing a single hidden layer, we accelerate the training process while simultaneously mitigating the risk of overfitting.

We evaluated the impact of the number of units in the hidden layer on classification accuracy. Figure 5b shows the results obtained in the SZ vs. HC task. When there are only ten units in the hidden layer, the accuracy reaches 92.87%. On the other hand, when the number of units exceeds 40, the accuracy stabilizes at a higher range of 97.85% to 97.89%. We observed that when the number of units in the hidden layer is 40, CDON achieves optimal performance, with a classification accuracy of 98.82%.

### 5.3. Parameters of Diffusion Module

The dimension of the higher-order representations stands as the most pivotal parameter in the Diffusion Module, and any alterations to it may exert a substantial impact on the performance of CDON-BD. In this subsection, we select the optimal dimension for higher-order representations based on accuracy. The experimental results in SZ vs. HC are illustrated in Figure 5b. The dimension ranges from 10 to 80 in increments of 10. The optimal dimension for higher-order representations is 60.

### 5.4. Various Latent Features

We also assess the influence of various latent features on the performance of CDON. Within CDON, the encoder and decoder play distinct roles in data compression and reconstruction, respectively [64]. In our method, we extract the encoder weights as the latent features (denoted as $W_{e}$). However, it is worth noting that the decoder weights $W_{d}$ can also capture network information. The encoder weights effectively encapsulate significant features in the input data, whereas the decoder weights retain information on how to remap these features back to the original input data, thereby encompassing partial information of the input data [65].

As can be seen from Figure 4, various latent features have a significant impact on the performance. When utilizing the decoder weights of CDON, the classification accuracy decreases by at least 6.6%. One possible reason is that the encoder weights are directly influenced by the input data, whereas the decoder is positioned later in the model structure, leading to information loss. This provides further evidence that the encoder weights excel in capturing features, whereas the decoder weights only capture partial features.

### 5.5. Effects of Modal

Although our method, which relies solely on fMRI data, achieves competitive performance, it has limitations compared to multimodal data fusion approaches. Multimodal methods can integrate complementary information from different sources, enhancing model generalizability [66]. For example, combining fMRI with diffusion tensor imaging (DTI) can incorporate functional and structural information, providing a more comprehensive understanding of the brain. As shown in Table 2, Multi-kernel SVM [35], HebrainGNN [29], MME-GCN [24], 3D-CNN [38], and OLFG [28] fuse fMRI with DTI, and Cascaded CNN [37] fuses fMRI with positron emission tomography (PET); their performance surpasses most methods that rely solely on fMRI, such as Autoencoder [23], Function Entropy SVM [36], H-FCN [21], nSEAL [27], and DCNs [22]. However, unimodal methods relying solely on fMRI have certain advantages, such as lower data acquisition and computational costs, which are significant in practical applications. Additionally, focusing on fMRI allows for in-depth exploration of brain functional patterns. Future work may involve integrating multimodal data to improve the performance and generalizability of our method.

### 5.6. Template Generalization

This study exclusively uses the AAL template for constructing brain networks without employing other templates such as Power264 [67] mainly because the AAL template is widely adopted in previous research [68,69,70]. In comparison, the Power264 template offers higher resolution with 264 regions, which can capture more detailed network connection changes and potentially improve the analysis of specific functional networks [71].

However, Wu et al. [72] recommend the AAL template over Power264 due to its higher reliability score. Additionally, Liu et al. [73] found that the AAL template slightly outperformed other templates like SC-100 [74] and BN-246 [75] in classifying mild cognitive impairment (MCI) and autism spectrum disorder (ASD), which suggests that the AAL template has strong generalizability across different conditions.

In future work, it is worth exploring the impact of various templates on classification performance to optimize brain network analysis methods.

### 5.7. Limitations

Our current study has a few limitations. Firstly, we relied exclusively on fMRI data, and incorporating multimodal data fusion could potentially enhance the performance. Secondly, we only used the AAL template for constructing brain networks without employing other templates. Future research will address these limitations to improve the performance and generalizability of our method.

## 6. Conclusions

In this paper, we propose the Continuous Dictionary of Nodes model and Bilinear-Diffusion (CDON-BD) network, aimed at automatically capturing and learning higher-order representations for brain disease analysis. We innovatively utilize the encoder weights of CDON to capture latent features, significantly enhancing its discriminability. The Bilinear Pooling technique fuses latent features and constructs higher-order brain networks. Based on it, the Diffusion Module learns higher-order representations from a global perspective for disease diagnosis. CDON-BD demonstrates excellent performance on two real datasets, effectively identifying regions associated with brain diseases and providing a novel perspective for comprehending disease pathology and supporting diagnostic endeavors.

## Figures and Tables

**Figure 1 brainsci-14-00810-f001:**
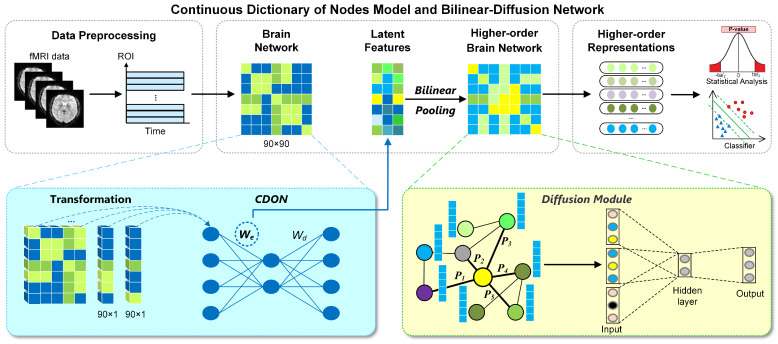
The architecture of the proposed CDON-BD network. The data preprocessing module extracts ROI time series from fMRI data and constructs the brain network. The CDON model captures latent features in the brain network through encoder weights. Bilinear Pooling further integrates latent features to construct higher-order brain networks. The Diffusion Module learns higher-order brain networks and generates higher-order representations, enabling statistical analysis and classification of brain diseases.

**Figure 2 brainsci-14-00810-f002:**
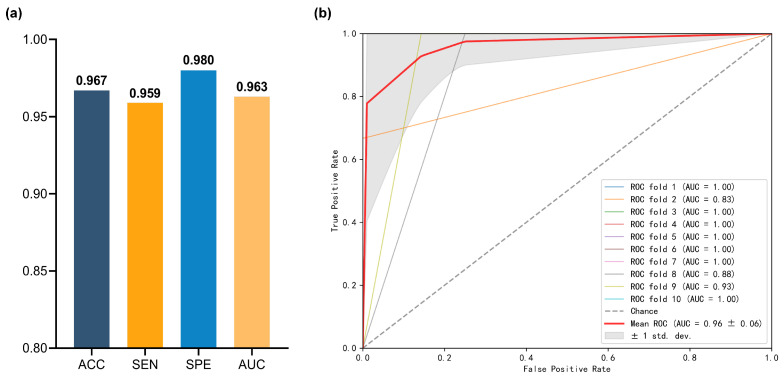
Classification performance in SZ vs. BD. (**a**) ACC, SEN, SPE, and AUC, (**b**) ROC curve.

**Figure 3 brainsci-14-00810-f003:**
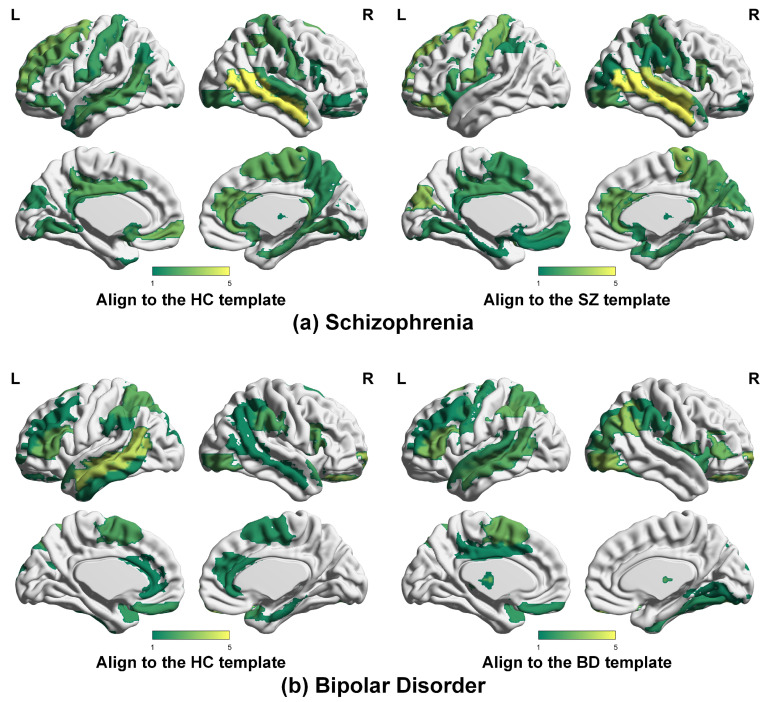
Brain regions with significant group differences (*p*-FDR < 0.05) between healthy controls and patients in SZ vs. HC and BD vs. HC.

**Figure 4 brainsci-14-00810-f004:**
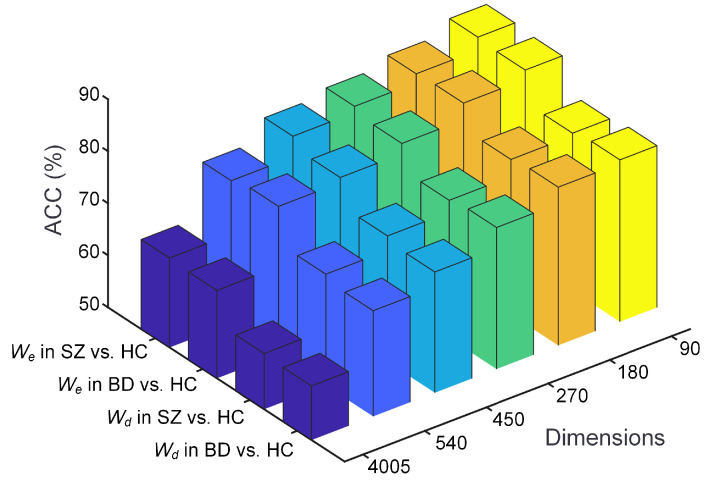
The impact of different input dimensions and latent features on the performance of CDON. $W_{e}$ and $W_{d}$ denote the latent features captured by the encoder and decoder, respectively.

**Figure 5 brainsci-14-00810-f005:**
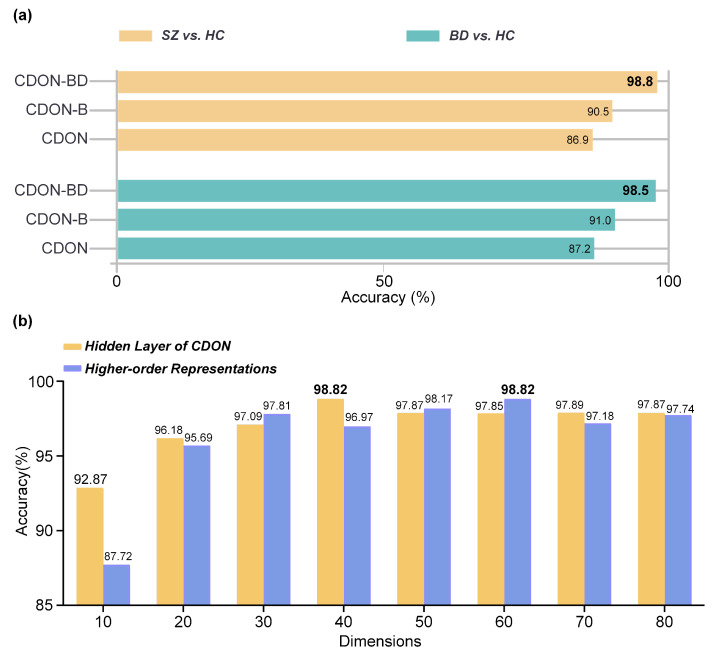
Ablation study of CDON, Bilinear Pooling, and Diffusion Module. (**a**) Effectiveness of the Bilinear Pooling and the Diffusion Module, (**b**) the effect of hidden layer dimensions and higher-order representation dimensions on the performance of CDON-BD in SZ vs. HC. Values in bold indicate the optimal performance.

**Table 1 brainsci-14-00810-t001:** Demographic information of participants.

Data	Age (Mean ± Std)	Gender (Female/Male)	Total
SZ	35.8 ± 8.7	14/34	48
BD	35.3 ± 8.9	21/28	49
HC	32.9 ± 8.2	20/30	50

SZ: schizophrenia; BD: bipolar disorder; HC: healthy controls.

**Table 2 brainsci-14-00810-t002:** Comparison of the classification performance (in %) between the SOTA methods and our method.

Method	SZ vs. HC				BD vs. HC			
	ACC	SEN	SPE	AUC	ACC	SEN	SPE	AUC
Autoencoder (2018) [23]	78.6	76.5	80.0	-	-	-	-	-
Function Entropy SVM (2023) [36]	79.2	-	-	-	87.5	-	-	-
H-FCN (2020) [21]	86.8	86.7	86.8	-	86.0	84.5	87.4	-
nSEAL (2020) [27]	87.5	84.2	88.5	0.863	88.9	92.1	85.4	0.888
DCNs (2018) [22]	90.5	89.7	91.5	0.908	91.6	91.2	92.0	0.919
Cascaded CNN (2018) [37]	95.2	96.8	93.6	-	95.0	96.5	93.4	-
Multi-kernel SVM (2019) [35]	95.6	94.2	97.1	0.959	95.8	96.5	95.1	0.960
HebrainGNN (2022) [29]	95.6	93.1	97.5	0.953	96.0	95.8	96.3	0.961
MME-GCN (2022) [24]	95.9	98.0	94.2	0.961	95.9	95.8	96.3	0.961
3D-CNN (2021) [38]	96.1	96.7	95.4	-	96.0	95.8	96.2	-
OLFG (2023) [28]	96.8	96.3	98.0	0.971	96.7	96.1	97.8	0.969
Ours	98.8	97.3	99.1	0.993	98.5	98.6	97.8	0.980

**Table 3 brainsci-14-00810-t003:** Comparison of the performance (in %) of different methods on the COBRE dataset.

Method	SZ vs. HC		
	ACC	SEN	SPE
Autoencoder (2018) [23]	73.6	72.0	74.5
nSEAL (2020) [27]	82.4	91.3	72.5
DNN (2016) [60]	85.8	86.3	85.3
MVRC (2017) [61]	89.0	-	-
3D CNN-LSTM (2023) [62]	92.3	93.9	88.8
E-RCN (2022) [63]	93.1	95.8	90.5
3D CNN (2019) [59]	98.1	97.5	98.6
CDON-BD	98.6	99.5	95.0

## Data Availability

The original data presented in the study are openly available at https://openfmri.org/dataset/ds000030/ (accessed on 18 January 2023) and https://fcon_1000.projects.nitrc.org/indi/retro/cobre.html (accessed on 19 January 2023).

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
