# Peer review of "Continuous Dictionary of Nodes Model and Bilinear-Diffusion Representation Learning for Brain Disease Analysis"

_brainsci, 2024, doi:10.3390/brainsci14080810_

Round 1

Reviewer 1 Report

Comments and Suggestions for Authors

Reviewer 2 Report

Comments and Suggestions for Authors

Continuous Dictionary of Nodes Model and Bilinear-Diffusion Representation Learning for Brain Disease Analysis

Introduction: In the section, the paragraph about studying brain networks with fMRI needs to be rewritten. Indeed, it is not clear whether the construction of brain networks depends by fMRI (lines 20-21).

The statement about the representation of learning methods needs more citation and clarified, since it is not clear if studies about node interaction did not exist(lines 38- 39). In this way, I suggest adding more references and clarifying this theoretical position.  Similarly, the random walks applied to neuroimaging data need to be explained in a better way, and explain the abstract vs low-level features.  I suggest putting the hypotheses before the methods and maybe in a different subsection.

Section 2. The statement “Recently, brain networks have been widely applied in brain disease analysis” needs to be rewritten. Brain networks are usually functional or structural networks present in the brain and could be altered by pathological conditions. Please, revise.

The concept of a “high-order network” needs to be explained in a better way. The section is however interesting, but it is not clear why the authors applied CDON-BD to the study of schizophrenia since few mention was made about it in the introduction. Schizophrenia and related brain alterations have been widely studied with different brain imaging techniques and different analytical techniques.  Please, introduce it. Moreover, the overlap between SZ and BD is not present in the introduction and needs to be discussed. A reader could ask “Why this specific comparison?”. The authors should add theoretical and previous evidence to fundament the study and the application of CDON-BD. This is an important and crucial theoretical issue.

Methods: add please the software that you used to perform the preprocessing.

The figure 1 is informative. The application of the bandpass filter and smoothing need to be motivated or references need to be added. The statement between 146 and 147 needs to be revised. It is not clear the use of the term neurons.

I advise that the generalizability of the method needs to be explained in the methods section.

The results are interesting, overall those obtained for Autoenconder accuracy, but the proposed model reached good results.

The discussion is in line with the results, and I agree with the authors that the absence of a direct comparison between SZ and BD is a limitation. However, this cannot justify the choice of these specific patient populations, that indeed was not completely stated in the main text. 

Round 2

Reviewer 1 Report

Comments and Suggestions for Authors

The manuscript is good and high level. I advice for publication. 

Author Response

Thank you very much for your positive feedback and for recommending our manuscript for publication. We appreciate your thoughtful comments and suggestions, which have been invaluable in refining our work.

Reviewer 2 Report

Comments and Suggestions for Authors

You performed the comparison between SZ and BD, and you performed a great job of revision.

Only a few commentaries:

“Brain networks constructed through various neuroimaging techniques can analyze the brain's neural activity and structural characteristics under different disease states” This statement is not clear in terms of construction.

“Many studies have found that the brain networks of SZ and BD may be altered by pathological conditions” SZ and BD are pathological conditions that show a series of cognitive, emotional and behavioral alterations. Several studies have found that SZ, and BD show network alterations when compared to HC. Please clarify. 

Author Response

Thank you very much for your positive feedback. We appreciate your comments and suggestions, which have been invaluable in refining our work.
